# Human Cytomegalovirus Reduces Endothelin-1 Expression in Both Endothelial and Vascular Smooth Muscle Cells

**DOI:** 10.3390/microorganisms9061137

**Published:** 2021-05-25

**Authors:** Koon-Chu Yaiw, Abdul-Aleem Mohammad, Chato Taher, Huanhuan Leah Cui, Helena Costa, Ourania N. Kostopoulou, Masany Jung, Alice Assinger, Vanessa Wilhelmi, Jiangning Yang, Klas Strååt, Afsar Rahbar, John Pernow, Cecilia Söderberg-Nauclér

**Affiliations:** 1Department of Medicine, Solna, Microbial Pathogenesis Unit, Karolinska Institutet, SE 171 64 Stockholm, Sweden; abdulaleem.research@gmail.com (A.-A.M.); chato.taher@gmail.com (C.T.); huancui@gmail.com (H.L.C.); hmarscosta@gmail.com (H.C.); masany.jung@gmail.com (M.J.); wilhelva@gmail.com (V.W.); afsar.rahbar@ki.se (A.R.); 2Division of Neurology, Karolinska University Hospital, SE 171 64 Stockholm, Sweden; 3Department of Oncology and Pathology, Karolinska Institutet, SE 171 64 Stockholm, Sweden; ourania.kostopoulou@ki.se; 4Institute of Vascular Biology and Thrombosis Research, Center for Physiology and Pharmacology, Medical University of Vienna, A-1090 Vienna, Austria; alice.assinger@meduniwien.ac.at; 5Department of Molecular Medicine and Surgery, Karolinska Institutet and University Hospital Solna, SE 171 64 Stockholm, Sweden; jiangning.yang@ki.se (J.Y.); John.Pernow@ki.se (J.P.); 6Department of Medicine, Division of Hematology, BioClinicum and Centre for Molecular Medicine, Karolinska University, Hospital Solna and Karolinska Institutet, SE 171 64 Stockholm, Sweden; klas.straat@ki.se

**Keywords:** human cytomegalovirus, immediate-early, endothelin-1, endothelial cells, smooth muscle cells

## Abstract

Human cytomegalovirus (HCMV) is an opportunistic pathogen that has been implicated in the pathogenesis of atherosclerosis. Endothelin-1 (ET-1), a potent vasoconstrictive peptide, is overexpressed and strongly associated with many vasculopathies. The main objective of this study was to investigate whether HCMV could affect ET-1 production. As such, both endothelial and smooth muscle cells, two primary cell types involved in the pathogenesis of atherosclerosis, were infected with HCMV in vitro and ET-1 mRNA and proteins were assessed by quantitative PCR assay, immunofluorescence staining and ELISA. HCMV infection significantly decreased ET-1 mRNA and secreted bioactive ET-1 levels from both cell types and promoted accumulation of the ET-1 precursor protein in infected endothelial cells. This was associated with inhibition of expression of the endothelin converting enzyme-1 (ECE-1), which cleaves the ET-1 precursor protein to mature ET-1. Ganciclovir treatment did not prevent the virus suppressive effects on ET-1 expression. Consistent with this observation we identified that the IE2-p86 protein predominantly modulated ET-1 expression. Whether the pronounced effects of HCMV in reducing ET-1 expression in vitro may lead to consequences for regulation of the vascular tone in vivo remains to be proven.

## 1. Introduction

Human cytomegalovirus (HCMV) belongs to the family *Herpesviridae* and subfamily *Betaherpesvirinae*. It is a ubiquitous but species-specific beta-herpesvirus that remains latent or persistent in certain host cells upon primary infection. Its seroprevalence varies from 45–100% depending on age, gender, socioeconomic status and geographical locations [1]. The viral replication cycle is regulated by a cascade of gene expression events. It is initiated from the major immediate-early (*MIE*) region, of which the main IE proteins IE1-p72 (*IE72*) and IE2-p86 (*IE86*) are transcription factors regulating early and late viral gene expression, as well as the expression of many cellular genes. The virus has 252 open reading frames believed to encode for about 200 proteins [2]. However, ribosomal profiling data suggest that over 750 unique viral mRNAs are being translated in infected cells [3], which suggests that this virus is much more complex than initially thought. Most of the CMV proteins are not required for viral replication and formation of new virus particles but are instead able to affect cellular functions and the immune system in ways that have helped the virus to co-exist with its host. These mechanisms could also promote various human pathologies [4,5].

HCMV is a leading cause of congenital infections with a substantial risk of birth defects. It also causes major illness and sometimes death in immunocompromised patients such as transplant patients and AIDS patients. Although its omni presentation in additional various pathologies precedes its definite pathogenic role, an active role of HCMV has also been proposed in various vasculopathies, chronic inflammatory conditions and in many types of cancers [5,6,7,8,9]. For instance, HCMV is associated with atherosclerosis, transplant vascular sclerosis, restenosis, abdominal aortic aneurysms and has been implied to play a role in type II diabetes, hypertension, and thrombotic events [10,11,12,13]. High levels of HCMV IgG antibodies are significantly related to increased mortality from cardiovascular disease [14,15] and are increased in patients with type II diabetes [16] and hypertension [17]. Virus proteins have been detected in heart transplants, in atherosclerotic plaques and in aortic aneurysms, but are rarely found in healthy vessels [13,18,19,20,21]. Emerging evidence also demonstrates a very high prevalence of HCMV proteins in several forms of cancer including glioblastoma, medulloblastoma, colon, prostate, and breast cancers [9,22,23,24,25]. While tumors are positive, healthy tissue around these tumors are virus negative. These observations imply that HCMV could be involved in the pathogenesis of a variety of diseases. Some studies have however failed to link HCMV to cancer or atherosclerosis [26,27,28,29], especially when non-optimized methods have been used for virus detection [30]. Definitive answers for a role of HCMV in a specific disease would come from treatment studies. In animal models CMV aggravates atherosclerosis and cancer growth and anti-CMV therapy reduces vascular disease as well as the growth of tumors [9,25,31,32,33]. Anti-viral prophylactics against HCMV prevent transplant vasculopathy or chronic rejection [34,35,36,37,38]. At present, six clinical drugs are available for HCMV treatment; these include ganciclovir (GCV) and its oral prodrug-valganciclovir, foscarnet, cidofovir, fomivirsena and letermovir. These are primarily used for treatment of severe HCMV infections but one, valganciclovir, is currently under our evaluation to determine if it can improve the outcome of glioblastoma patients, whose tumors are almost always HCMV positive (NCT04116411).

Endothelin-1 (ET-1), the most abundant protein of the endothelin family (ET-1, ET-2 and ET-3), is a 21-amino-acid long potent vasoconstrictor peptide that plays a pathophysiological role in various cardiovascular diseases including pulmonary arterial hypertension and atherosclerosis, as well as non-cardiovascular inflammatory diseases such as inflammatory bowel disease [39,40], rheumatoid arthritis [41,42,43] and SLE [44]. The endothelin axis is constituted by ET-1 and its receptors (either endothelin A (ETAR) or B (ETBR) receptor) and plays an important modulatory role in development, cell proliferation, apoptosis and immune responses and has been proposed to contribute to the growth and progression of many tumor types (reviewed in [45,46,47,48,49]).

The bioavailability of ET-1 is mainly regulated at the transcriptional level and little or no ET-1 is stored in the granules of ET-producing cells (reviewed in [46,50]). The human edn1 gene consists of 5 exons and transcribes a 2.8-kb mRNA encoding preproET-1 of 212-aa. The mature ET-1 peptide is produced through a cascade of proteolytic processes beginning from the conversion of preproET-1 into a 38-aa proET (also referred to as big ET-1) by a furin-like endopeptidase before being further cleaved by the ET converting enzymes (ECE) into the mature, biologically active 21-aa ET-1 peptide (reviewed in [46,49,50]).

Since expression of HCMV proteins and ET-1 are implied in various cardiovascular diseases as well as in many cancers, we hypothesized that there may be an interplay between HCMV and ET-1 that may be of relevance to understand HCMVs role in the pathogenesis of these diseases. We therefore investigated whether HCMV can affect the expression of ET-1 in human primary endothelial and smooth muscle cells. We found that HCMV downregulates ET-1 expression and secretion mainly via the action of its IE2-p86 protein, with a simultaneous accumulation of the ET-1 precursor protein preproET-1 in infected cells. This was associated with an inhibition of ECE-1, which had direct consequences on the expression of an isoform of the HCMV IE protein. These observations may help us to further understand the role of HCMV in vascular diseases and cancer.

## 2. Materials and Methods

### 2.1. Virus Stock and Cells

The clinical HCMV strain we used for the study, VR 1814, was a kind gift from Dr. G. Gerna, University of Pavia, Italy [51]. The virus was propagated in human umbilical vein endothelia cells (HUVEC) and the extracellular free virus in the supernatant was either directly used as virus stock or pelleted by ultracentrifugation as previously described [52,53] and used as a virus concentrate to infect cells. Virus titration was performed in fibroblast cells, MRC-5 as previously described [54]. The cells used in this study were HUVEC either from ATCC (cat no. CC-2517, ATCC, Manassas, VA, USA) or HUVEC isolated as earlier described from anonymized umbilical cords collected from Karolinska University Hospital [52,55]. Ethical approval for endothelial cell isolation and subsequent experimentation was granted by the Regional ethical committee in Stockholm, No 2015/1294-31/2). HUVECs were used between passage no. 3–5, human pulmonary aortic endothelial cells (hPAEC, cat no. CC-2530; used between passage no. 6–8), human pulmonary arterial smooth muscle cells (hPASMC, cat no. CC-2581; used between passage no. 4–6) and human aortic smooth muscle cells (hASMC; between passage no. 6–8), fibroblasts MRC-5 (cat no. CCL-171 and used between passage no. 16–20) cells. The growth conditions for cells were as previously described [53]. The HUVEC, hPAEC and hPASMC cells were purchased from Clonetics, Lonza and MRC-5 cells were from ATCC. The hASMC stock was from the previous study [13]. All cells were verified Mycoplasma free by MycoAlert mycoplasma detection kit (cat no. LT07-318, Clonetics, Lonza, Basel, Switzerland).

### 2.2. Virus Infection

Cells were infected with different multiplicity of infection (MOI) of HCMV when cells had reached a confluence of about 80–85% in 12-well plates. To account for the different volume due to different MOI, the viral inoculum was standardized to a final volume of 1 mL with growth medium. After a 2-h virus pre-adsorption at 37 °C, 1 mL of fresh medium was then added to each well. The cells and supernatant were harvested at 1-, 3- and 5-day post-infection (dpi) and in some experiments at 3-, 6-, 18-h before being subjected to total RNA and protein extraction/isolation or enzyme-linked immunosorbent assay (ELISA), respectively. The UV-irradiated HCMV was originated from the same virus stock used for infection experiment and was irradiated using Stratagene Stratalinker^®^ UV Crosslinker 1800 as described previously [53]. The same MOI of UV-irradiated HCMV was then used in parallel with the infection experiment. For GCV treatment, HUVEC were infected together with 1-, 10- and 50 µM of GCV (cymevene, Roche, Basel, Switzerland) for 5 days prior to samples being collected for analysis with a TaqMan real-time PCR assay.

### 2.3. RNA Isolation and TaqMan Real-Time PCR Assay

The total RNA isolation, cDNA synthesis and TaqMan real-time PCR assay were performed as previously described [53] with the following TaqMan primers/probe from the Applied Biosystems: ET-1 (assay ID, Hs00174961_m1), beta 2 microglobulin (B2M, assay ID, Hs00984230_m1) or 18S rRNA (cat no. 4333760) were used for normalization where appropriate. The primers target exon 2 and 3 boundary of the edn1 gene and the probe spans the exons, producing a 62-bp amplicon. The standard method with standard curve generated from ten-fold dilution of the pooled cDNA from each sample or the comparative delta delta CT method (2^−ΔΔCt^) method described in User Bulletin #2 ABI PRISM 7700 Sequence Detection System was used to quantify the gene expression.

### 2.4. Human Endothelin-1 QuantiGlo ELISA Kit

The ET-1 ELISA was performed according to the manufacturer’s instruction (R&D Systems, cat no. QET00B). In brief, 100 µL of culture media supernatant was added onto coated 96-well plate after addition of assay diluent and incubated for 1.5-h at RT. The plates were then washed prior to addition of conjugate for 3-h before being washed again and developed with working Glo reagent. The relative light unit was determined using Wallac Victor 1420 multichannel counter (PerkinElmer, Waltham, MA, USA) with the recommended settings.

### 2.5. Immunofluorescent Staining (IF)

The IF was performed on 10% neutral-buffered formalin-fixed cells in 8-chamber slides. The fixed cells were then permeabilized with 0.1% Triton-X 100 in PBS for 15 min at RT, washed twice before blocking with protein blocker (Dako) for 20 min, Fc receptor blocker (Innovec) for 15 min followed by 30 min incubation with normal horse serum (Vector). Primary antibody was mouse anti-IE (Millipore, cat. no. MAB810R, 1:500) and rabbit anti-ET-1 (Abcam, cat. no. ab170544, 1:100; targeting the preproET-1, 183–211 aa). For dual-immunofluorescent staining, both primary antibodies were added together and incubated at 4 °C for 16–18-h. Secondary antibody goat anti-mouse Alexa Fluor 488 (cat. no. A11001) and goat anti-rabbit Alexa Fluor 594 (cat. no. A11012) were added together and incubated for 1-h at RT. All the secondary antibodies were from the Molecular Probes, Invitrogen and used at 1:500 dilutions. Nuclei were counterstained with 4′-6-Diamidino-2-phenylindole provided with the mounting medium (Vector, H-1200) and slides were viewed under Leica confocal microscope and analyzed with Leica Application Suite Advanced Fluorescence software or Zeiss LSM 700 Confocal with Zen software.

### 2.6. Western Blot Analysis

The protein extraction and Western blot assay were essentially performed as previously described with the primary antibody mouse anti-IE (1:5000, Argene, France, cat. no. 11-003), mouse anti-β actin (1:1000, Santa Cruz, Dallas, TX, USA, cat. no.sc-47778), rabbit anti-ET-1 (1:200, Abcam, Cambridge, UK, cat. no. ab170544) and mouse anti-ECE-1 (A-6, 1:250, Santa Cruz, cat. No. sc-376017). Secondary antibody anti-mouse (1:20,000, cat. no. A9044) or anti-rabbit (1:10,000, cat. no. A0545) conjugated to horseradish peroxidase (both from Sigma-Aldrich, St. Louis, MO, USA) together with ECL-Prime chemiluminescence (Amersham, UK) that was used for detection.

### 2.7. Short Interfering RNA (siRNA) against IE72 and IE86

The siRNA sequence for IE72 (IE233) or IE86 and the transfection protocol using lipofectamine 2000 were as described previously [53,54]. In brief, 80–90% of confluent HUVEC in 12-well plates were first transfected with siRNA for 24-h prior to infection with HCMV. The cells were then harvested for analysis with a TaqMan real-time PCR assay after 96-h of infection.

### 2.8. Statistical Analysis

Results were expressed as mean ± standard deviation (SD) and analyzed with an unpaired t-test with two-tailed *p* value and 95% confidence interval incorporated within the GraphPad Prism software version 6. A *p*-value < 0.05 was considered statistically significant.

## 3. Results

### 3.1. HCMV Infection Downregulates ET-1 Transcript Production and Release of ET-1 Peptide from Infected Endothelial and Smooth Muscle Cells

To investigate the effects of HCMV infection on ET-1 mRNA expression, we infected endothelial cells (HUVEC) and smooth muscle cells (hPASMC) with HCMV at different multiplicity of infection (MOI) of HCMV and harvested the cells at 1-, 3- and 5-dpi. HCMV infection unequivocally and strongly downregulated ET-1 mRNA expression in a time and dose-dependent manner in both cell types (Figure 1A,B). As expected, the HCMV MIE expression increased over time after infection and was MOI-dependent in both cell types (Figure 1C,D).

Similar suppressive effects on ET-1 mRNA expression were also observed in another HCMV-infected endothelial cell, hPAEC and in smooth muscle cells, hSMC (Figure 2A,B). As expected, the MIE expression increased over time after infection and was MOI-dependent in both cell types (Figure 2C,D).

Since ET-1 regulation mainly occurs at the transcriptional level and the peptide is released directly upon synthesis, we next quantified the secreted ET-1 in supernatants of HCMV-infected cells. Consistent with the reduction of ET-1 transcripts, the secreted ET-1 was also significantly reduced in supernatants from both infected endothelial and vascular smooth muscles cells, with the most pronounced effect at 5 dpi (Figure 3A–D).

As the ET-1 peptide is cleaved off from the preproET-1 protein, we next used an antibody targeting aa 183–211 of the preproET-1 to examine the expression levels of the preproET-1 protein and stained uninfected- and HCMV-infected endothelial cells to assess for protein expression. In sharp contrast to the significantly reduced expression levels of ET-1 transcripts and reduced release of the ET-1 peptide, we found a dose-dependent accumulation of the preproET-1 protein levels in HCMV infected cells (Figure 4A–C). To ascertain whether the preproET-1 protein level was upregulated by HCMV infection, we also performed a Western blot assay. As shown in Figure 4D, the ET-1 precursor protein, preproET-1 is accumulated in endothelial cells infected at a high MOI (MOI 10). A greater cytopathic effect was observed at later time-points of infection (Figure 4C) and was more pronounced in the highest MOI (MOI = 10), but cells did not lyse to the extent usually observed in infected fibroblasts. We also noted that increased preproET-1 expression was associated with expression of the HCMV IE86 protein (Figure 4D).

To further examine if HCMV induces accumulation of ET-1 precursor protein via an effect on ECE-1 (the enzyme responsible for cleaving the preproET-1 into its mature active form ET-1), we examined if HCMV infection affected expression levels of the ECE-1 protein in HCMV-infected HUVECs. We found that HCMV infection reduced ECE-1 expression levels in a dose-dependent manner (Figure 4E). At an MOI of 10, HCMV infection completely abolished detectable levels of ECE-1 protein expression. Concomitantly, we observed an accumulation of an immunoreactive high molecular weight IE protein (Figure 4E) in size range of 150–250 kDa. These observations imply a possible relationship between ECE-1 and processing of HCMV IE proteins.

### 3.2. HCMV IE86 Downregulates ET-1 Transcript Levels

The HCMV-IE proteins are the first to be transcribed during infection and are not dependent on viral DNA replication. As noted in Figure 5A,B, downregulation of ET-1 mRNA was evident already at 3-hpi in HUVEC cells and at 6-hpi in hSMC cells with a potential relationship to HCMV IE expression (Figure 5A,B). We detected viral IE expression at 3-hpi that peaked at 18-hpi before decreasing at 24-hpi in both cell types (Figure 5C,D).

To address the question whether HCMV’s effect on ET-1 mRNA levels requires active viral replication, we infected both endothelial and smooth muscle cells with UV-irradiated HCMV in the same manner as with non-UV treated virus. We detected very low levels of IE mRNA in UV-irradiated HCMV treated cells (threshold cycle of >35 at MOI = 10) in both HUVEC and hPASMC, while cells treated with lower MOIs were completely negative. UV treatment thus nearly completely abolished viral replication. We found that treatment of cells with UV-irradiated HCMV also downregulated ET-1 mRNA at 1- and 3-dpi in HUVEC (Figure 6A, MOI = 10), while this replicative deficient UV-treated virus did not affect expression of ET-1 at 5 dpi (Figure 6A). Downregulation of ET-1 mRNA was also observed in hPASMC at 3- and 5 dpi, although levels of ET-1 transcripts were also low at the later time point in uninfected cells (Figure 6B). Similarly, UV-treated virus reduced ET-1 expression in hSMC at 3 dpi (Figure 6C). Hence, a replication deficient virus could modulate ET-1 mRNA expression albeit the ET-1 downregulation was more pronounced in virus replicating cells.

### 3.3. Ganciclovir Does Not Prevent HCMV’s Inhibitory Effect on ET-1 Production, Which Is Mainly Regulated by HCMV IE2-p86

The main stay treatment for HCMV infection in patients is ganciclovir or its prodrug valganciclovir. This drug acts as a nucleoside analogue to prevent HCMV DNA replication and expression of late HCMV proteins but has little or no effect on HCMV IE protein expression. We therefore next investigated whether ganciclovir could prevent the viral effects on ET-1. To address this, we infected and treated the endothelial cells with GCV and quantified the ET-1 mRNA levels at 5 dpi. Even though GCV treatment significantly reduced IE mRNA expression, it did not prevent the virus mediated downregulation of ET-1 mRNA expression even at the highest dose tested (Figure 7A,B). This finding suggests that the viral IE gene products or other viral gene products produced immediately upon virus entry may mediate the inhibitory effects on ET-1 expression [56].

Both IE1-p72 (IE72) and IE2-p86 (IE86) are transcription factors acting to control expression of both viral and cellular gene expression and these are expressed in HCMV-infected GCV-treated cells. Since ET-1 was mainly regulated on a transcriptional level, we examined if either IE72 or IE86 affected ET-1 expression. To this end, we transfected HUVEC with siRNA against IE72 or IE86 prior to HCMV infection and harvested cells for mRNA analysis at 4 dpi. While silencing of IE72 reduced approximately 80% of its expression, it failed to prevent the HCMV-mediated ET-1 downregulation (Figure 8A,B). However, siRNA against IE86 distinctly restored the ET-1 mRNA levels even at about 50% silencing efficiency rate (Figure 8C,D). These observations imply that IE86, but not IE72 affects transcription of ET-1 (Figure 4E).

## 4. Discussion

ET-1 plays important roles in vascular diseases, as well as being a modulator of development, cell proliferation, apoptosis, immune responses and cancer [45,47,50,57,58,59,60]. Here, we demonstrate that HCMV has developed mechanisms to target the ET-1 axis, which imply a potential important role of ET-1 and its receptors ETAR and ETBR in the life cycle of this virus. This new insight into the virus’ effects on vascular cells in the early phase of HCMV infection will increase our understanding of HCMV pathogenesis.

ET-1 expression was targeted by HCMV and involved at least two mechanisms. We show that HCMV directly downregulated ET-1 mRNA expression and the release of ET-1 peptide from both human primary endothelial and smooth muscle cells. HCMV IE86 reduced transcript levels of ET-1 early after infection, and HCMV also downregulated ECE-1 expression which resulted in an accumulation of the ET-1 precursor protein prepro-ET-1, and reduced levels of the mature ET-1 peptide in the supernatant of infected cells when IE86 was expressed. The accumulation of preproET-1 may involve a negative feedback on this axis to further reduce release of ET-1. The viral effects on ET-1 mRNA occurred early after infection and were more pronounced with active viral infection, and this is consistent with the known role of IE86 to sometimes act as a transcriptional repressor [61].

In vascular diseases, elevated ET-1 contributes to endothelial dysfunction, atherosclerosis and vasculopathies such as pulmonary arterial hypertension (PAH). Bosentan and macitentan are dual ETAR/ETBR inhibitors that are currently for treatment of PAH to reduce the vascular tone and lower the blood pressure in these patients [62,63]. Here, we consistently observed that HCMV infection reduced ET-1 levels in vitro in both EC and SMC, which may affect the vascular tone during acute infection in vivo. As ET-1 is a vasoconstrictive peptide, one would, in such scenarios, expect less vascular contraction and reduced blood pressure. Studies of the role of HCMV in hypertension have instead shown that patients with hypertension have higher HCMV specific antibody levels. One could speculate that those with higher IgG levels to HCMV may have a better immune control of HCMV, and are perhaps then less affected by the virus infection. More important would be studies of blood pressure levels during an acute HCMV infection. About 30% of critically ill patients reactivate HCMV, which increases their mortality risk [64]. No specific studies have to our knowledge studied blood pressure levels during acute HCMV disease or when critically ill patients reactivate HCMV. A further complication is also given by the fact that that the virus affects the renin-angiotensin system which will also affect blood pressure levels [65]. It is possible that these mechanisms are interrelated from a viral evolutionary perspective.

Other viruses such as paramyxoviruses measles virus, canine distemper virus and influenza virus can also modulate the ET-1 system, and ET-1 has been suggested to play a role in viral persistence [66,67]. For example, in Vero cells chronically infected with influenza virus, preproET-1 mRNA and ET-1 secretion were suppressed [66]. In contrast, HIV induces upregulation of ET-1, most likely through its Tat protein and glycoprotein gp120 [68,69]. To our knowledge, this is the first report that a herpesvirus can reduce bioactive ET-1 mRNA expression and its secretion. One earlier report demonstrated that infection of human microvascular endothelial cells with the Kaposi sarcoma herpes virus (KSHV) did not affect the bioactive ET-1 but increased big ET-1 secretion at 3-days post infection [70]. Strikingly, we observed that the preproET-1 protein levels were accumulating in HCMV-infected endothelial cells in a dose-dependent manner, while both ET-1 transcripts and release of ET-1 peptides in the culture supernatants were reduced. The HCMV-induced accumulation of the preproET-1 occured concomitantly with the reduced levels of the ECE-1 protein levels, and this resulted in the accumulation of an unknown HCMV IE-immunoreactive protein(s) of high molecular weight (150–200 kDa in size). The significance of this finding is at present unknown but will be interesting to study further.

We recently showed a link between the ET-1 receptor ETBR expression and glioblastoma [70]. We also found that increased ETBR expression was associated with poor prognosis of glioblastoma patients [71]. We and others have earlier shown that the majority of glioblastoma tumors are HCMV positive [22,72,73], and that the HCMV IE protein levels are prognostic for patient survival [74]. We further found that the ET-1 receptor ETBR appears to play an important role in the life cycle of HCMV, by serving as a receptor for HCMV to promote viral entry and replication (unpublished observations). Therefore, HCMV and the endothelin axis may be important in the biology of glioblastoma. Thus, the endothelin axis plays a hitherto unknown but likely crucial role in the biology of HCMV. This may be driven by a competitive situation between ET-1 and HCMV for ETBR activation, and under such scenario HCMV likely benefits from reduced ET-1 levels. These observations imply a potential interference between HCMV, and the ET-1 axis.

In addition to the direct interference between HCMV, ET-1 and ETBR, previous studies show that HCMV during its evolution had a strong incentive to develop immune evasion mechanisms. In this context it is interesting to note that both the function and survival of dendritic cells have been suggested to be dependent on the endothelin axis [75]. We therefore speculate that HCMV-induced downregulation of ET-1 may dampen the antigen-presenting capacity of dendritic cells and macrophages and impair their ability to stimulate T cells and mount efficient immune responses. This may provide another immune evasion strategy employed by HCMV. Hence, further studies should be tailored to understand the effect of preproET-1 accumulation and ET-1 activity on HCMVs life cycle, and its impact on regulation of ETBR function and the immune system.

## 5. Conclusions

In conclusion, we show that HCMV strongly suppresses ET-1 by the action of its IE86 protein on ET-1 transcription and via inhibition of ECE-1 protein expression to inhibit cleavage of the preproET-1 protein into its mature active form of ET-1. From an evolutionary perspective, this virus may have targeted the ET-1 axis in different ways to reduce ET-1 levels released from infected cells in the vasculature with potential consequences for viral pathogenesis. HCMV has effects on the endothelin axis and hence may impact on several human diseases.

## Figures and Tables

**Figure 1 microorganisms-09-01137-f001:**
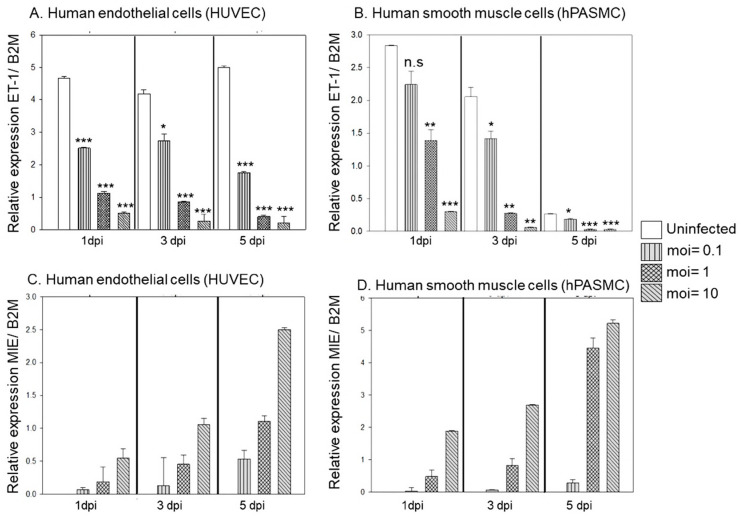
ET-1 mRNA expression was significantly downregulated in both HCMV infected HUVEC (**A**) and hPASMC (**B**) in a time- and dose-dependent manner, while HCMV-MIE transcript expression was upregulated, respectively (**C**) and (**D**). (Biological replicates, *n* = 3; DPI, days post infection; B2M, beta 2 microglobulin; values are mean ± SD. Significant levels compared to that of ‘Uninfected’: * *p* = 0.01–0.05; ** *p* = 0.001–0.01; *** *p* < 0.001; n.s. = not significant).

**Figure 2 microorganisms-09-01137-f002:**
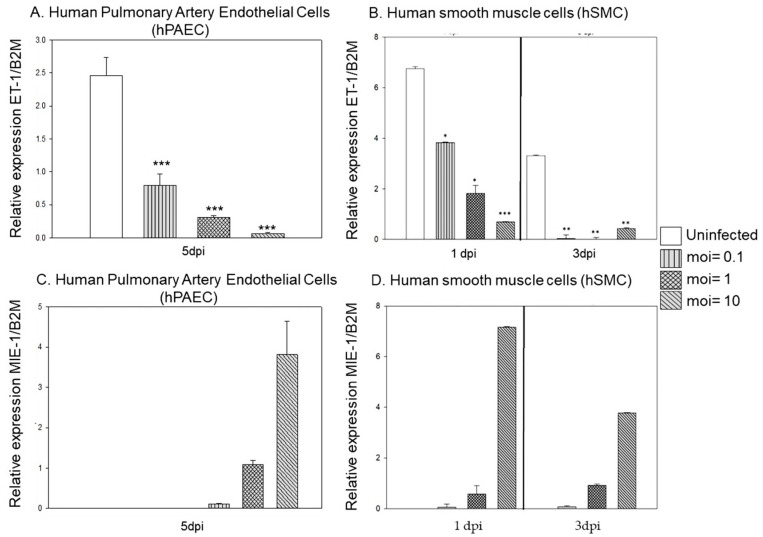
mRNA expression was significantly downregulated in the HCMV infected hPAEC at 5-dpi (**A**) and hSMC at 1- and 3-dpi (**B**), while (**C**,**D**) show the correspondent HCMV-MIE transcript expression. (Biological replicate, *n* = 2; DPI, days post infection; B2M, beta 2 microglobulin; values are mean ± SD. Significant levels compared to that of ‘Uninfected’: * *p* = 0.01–0.05; ** *p* = 0.001–0.01; *** *p* < 0.001).

**Figure 3 microorganisms-09-01137-f003:**
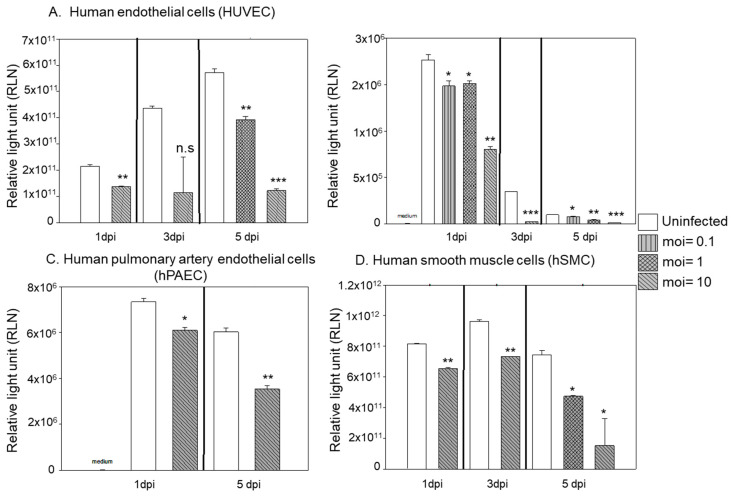
Secreted ET-1 protein levels were decreased in the supernatant of HCMV infected cells as quantified by ELISA and expressed by relative light unit. Protein levels of ET-1 were significantly decreased over time and in dose-dependent manner in supernatant of HCMV infected HUVEC (**A**), hPASMC (**B**), hPAEC (**C**) and hSMC (**D**). (Biological replicate, *n* = 2 for each cell type; HCMV, human cytomegalovirus; DPI, days post infection; MOI, multiplicity of infection; values are mean ± SD. Significant levels compared to that of ‘Uninfected’: * *p* = 0.01–0.05; ** *p* = 0.001–0.01; *** *p* <0.001; n.s. = not significant).

**Figure 4 microorganisms-09-01137-f004:**
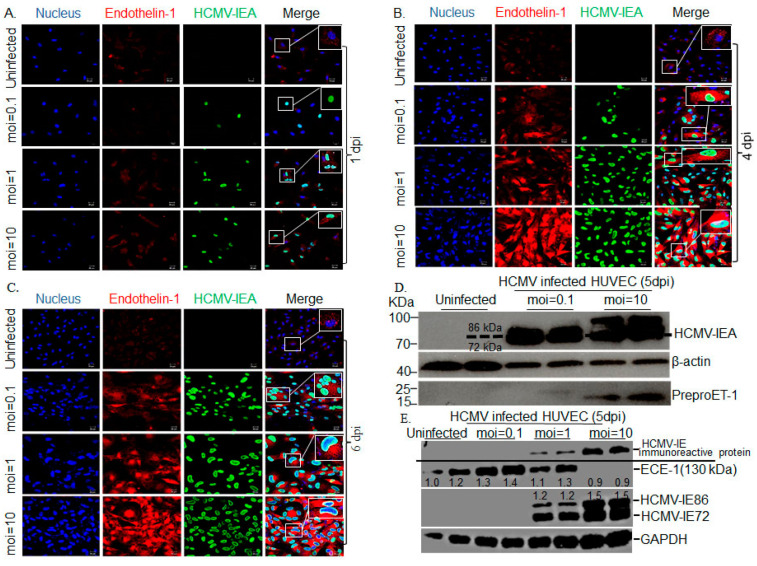
Increased accumulation of ET-1 protein is time- and MOI-dependent in HCMV-infected endothelial cells (HUVEC) as illustrated by representative images of immunofluorescence staining (**A**–**C**) and Western blot assay (**D**) 5 dpi. A dose-dependent accumulation of the preproET-1 protein levels in HCMV-infected cells over time ((**A**) (1 dpi), (**B**) (4 dpi), (**C**) (6 dpi)). The ET-1 protein levels of preproET-1 were accumulated in HCMV-infected HUVECs at MOI 10 and increased pre-proteinET-1 expression was associated with expression of the HCMV IE86 protein (**D**). The ECE-1 expression level was reduced in HCMV-infected HUVEC in a dose-dependent manner ((**E**) 5 dpi)) (Biological replicate, *n* = 3 or *n* = 2 for immunofluorescence or Western blot (representing 2 lanes for each), respectively; HCMV, human cytomegalovirus; DPI, days post infection; MOI, multiplicity of infection; IEA, immediate early proteins; ECE, endothelin converting enzyme; GAPDH, glyceraldehyde 3-phosphate dehydrogenase).

**Figure 5 microorganisms-09-01137-f005:**
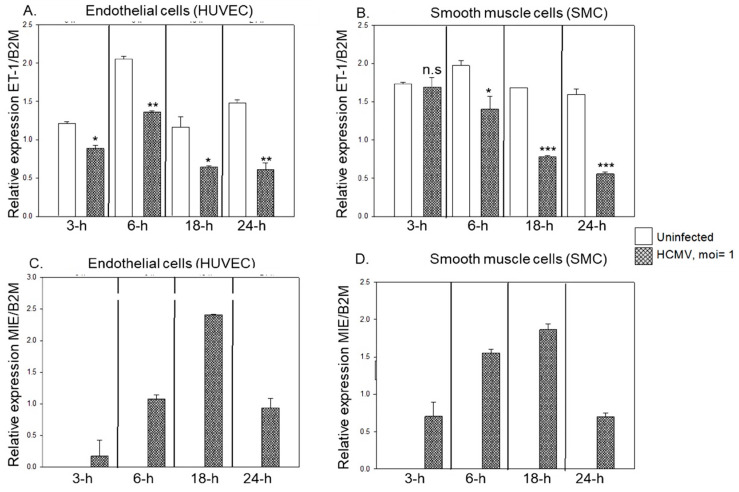
HCMV-IE downregulates ET-1 transcription. ET-1 mRNA was significantly downregulated at 3-, 6-, 18- and 24-hpi in HUVEC cells and at 6-, 18- and 24-hpi in hSMC (**A**,**B**). HCMV-IE was expressed at 3-h after infection and peaked at 18-h before decreasing at 24-hpi in both cell types (**C**,**D**). (Biological replicates, *n* = 2 for each cell type; HCMV, human cytomegalovirus; H, hours; ET, endothelin-1; MIE, major immediate early; B2M, beta 2 microglobulin; values are mean ± SD. Significant levels compared to that of ‘Uninfected’; * *p* = 0.01–0.05; ** *p* = 0.001–0.01; *** *p* < 0.001; n.s. = not significant).

**Figure 6 microorganisms-09-01137-f006:**
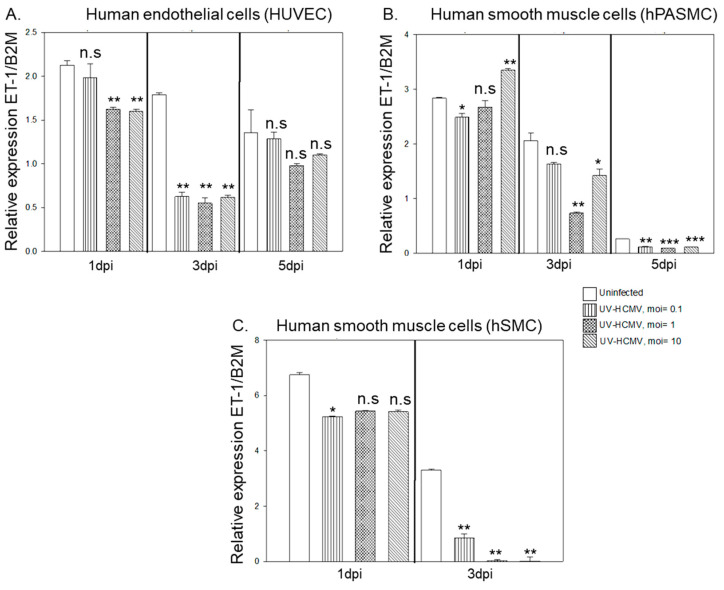
Replicative deficient virus can modulate ET-1 mRNA expression. UV irradiation of HCMV results in downregulation of ET-1 mRNA at 1- and 3-dpi in both endothelial and smooth muscle cells (**A**–**C**). (Biological replicate, *n* = 3 for HUVEC and hPASMC, *n* = 2 for hSMC; HCMV, human cytomegalovirus; UV, ultraviolet light; DPI, days post infection; MOI, multiplicity of infection; ET-1, endothelin-1; B2M, beta 2 microglobulin; values are mean ± SD. Significant levels compared to that of ‘Uninfected’; * *p* = 0.01–0.05; ** *p* = 0.001–0.01; *** *p* < 0.001; n.s. = not significant).

**Figure 7 microorganisms-09-01137-f007:**
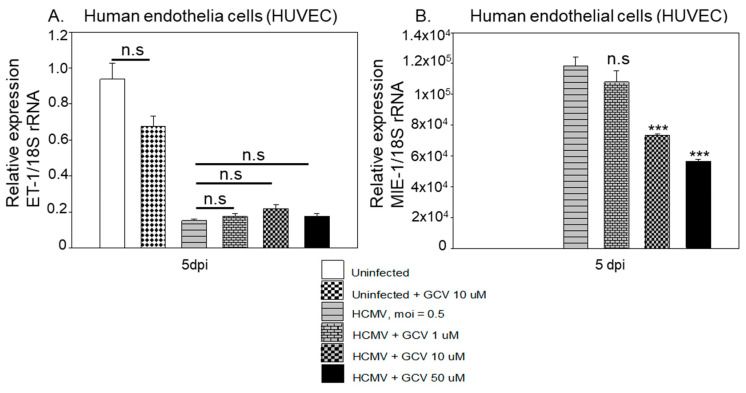
Ganciclovir does not prevent HCMV’s inhibitory effect on ET-1 production in HUVEC. GCV treatment of HCMV infected cells did not prevent the virus-mediated downregulation of ET-1 mRNA expression (**A**), but unexpectedly also significantly reduced IE mRNA expression (**B**). (Biological replicates, *n* = 3; HCMV, human cytomegalovirus; DPI, days post infection; MOI, multiplicity of infection; ET-1, endothelin-1; MIE, major immediate early; rRNA, ribosome ribonucleic acids; values are mean ± SD. Significant levels compared to that of ‘Uninfected’; *** *p* < 0.001; n.s. = not significant).

**Figure 8 microorganisms-09-01137-f008:**
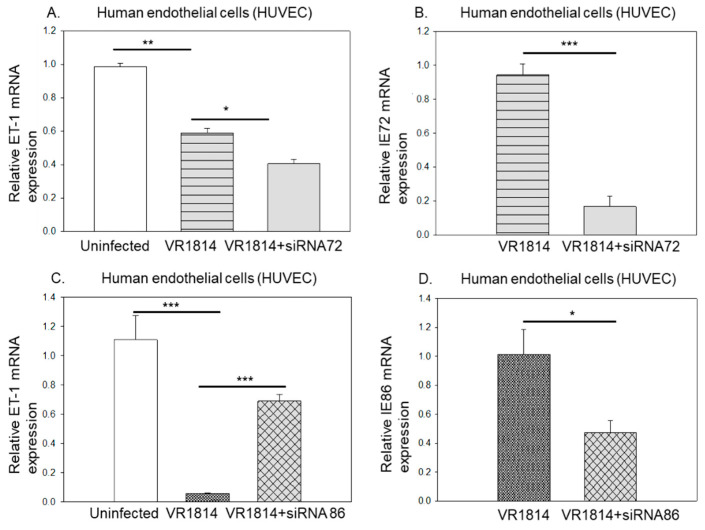
HCMV IE86 controls expression of ET-1 mRNA. The effects of siRNA silencing HCMV IE72 and IE86, respectively, on the expression of ET-1 transcription in HUVECs (**A** and **C**). Silencing of IE72 reduced expression of IE72 by approximately 80% but did not prevent the HCMV-mediated ET-1 downregulation (**A**,**B**). siRNA against IE86 specifically restored the ET-1 mRNA levels even at about 50% silencing efficiency rate (**C**,**D**). (Biological replicates, *n* = 3; HCMV, human cytomegalovirus; DPI, days post infection; MOI, multiplicity of infection; ET-1, endothelin-1; IE, viral immediate early; values are mean ± SD. Significant levels compared to that of ‘Uninfected’; * *p* = 0.01–0.05; ** *p* = 0.001–0.01; *** *p* < 0.001; n.s. = not significant).

## Data Availability

The data presented in this study are available on request from the corresponding authors.

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
