# Peer review of "Human Cytomegalovirus Reduces Endothelin-1 Expression in Both Endothelial and Vascular Smooth Muscle Cells"

_microorganisms, 2021, doi:10.3390/microorganisms9061137_

Round 1
Reviewer 1 Report
The authors present a well planned and clear study showing down regulation of ET-1 secretion following in vitro infection.
Minor issues
The Figure legends need further details and greater consistency – this can be achieved easily. It may also be worth looking at the fonts used in the graphs. The * and ** are not defined.
The last paragraph of the discussion mentions dendritic cells. This is not the topic of the present paper.
Major issues
The higher MOI used here resulted in the infection of 100% of the cells. In vivo severe infections do not achieve 1% infection. Accordingly, there is a strong precedent in the mouse literature for conflicting effects in vivo & in vitro. Hence there is a need in an individual culture to distinguish between cells infected and cells affected by HCMV. At the least this should be discussed as a limitation.
On this theme, it would also be helpful to see the immunohistochemistry at a higher magnification and with a clearer description of which cells were infected and affected. Were the infected cells altered morphologically? Eg: rounded up and non-functional by other assessments?
Technical issues requiring clarification
B2M and 18S RNA seem to be used interchangeably to standardize the mRNA assays. Were levels of these 2 mRNAs directly related?
The ELISAs were recorded as OD…. Can we be sure all measurements were made in the linear portion of the curve?
How was MOI determined? In what cell type?
The statistical analysis is described simply as mean+SD. Is that based on replicate wells (how many) in a single experiment…or mean values derived from several experiments?
Author Response
|
Department of Medicine Karolinska Institutet |
Stockholm, May 19th , 2021 |
Dear Editor
Thank you for your interest in our work, and for your willingness to consider acceptance of our manuscript entitled " Human cytomegalovirus reduces endothelin-1 expression in both endothelial and vascular smooth muscle cells” for publication in the Microorganism, after we have carried out the corrections as suggested by the reviewers.
We thank the reviewers for their valuable comments, which we think improved the quality of our manuscript. We have addressed the reviewers comments below in the point-by-point response and rewritten the manuscript accordingly. We hope that our manuscript now can be accepted for publication in the Microorganisms.
All authors concur with the resubmission of this manuscript. We are willing to pay the cost of printing the figure in colour. If I can be of any further assistance in this matter, please do not hesitate to contact me at your earliest convenience.
Yours Sincerely,
Professor Cecilia Söderberg-Nauclér
Institutet, Sweden
On behalf of the authors
Point-by-point response:
Reviewer 1
Minor issues
The Figure legends need further details and greater consistency – this can be achieved easily. It may also be worth looking at the fonts used in the graphs. The * and ** are not defined.
A: We have added relevant information for the Figure legends and tried to keep the fonts as consistent as possible. In addition, we have defined the statistics significant levels where appropriate (see the revised Figures and respective legends in the revised ms).
The last paragraph of the discussion mentions dendritic cells. This is not the topic of the present paper.
A: We thank the reviewer for the comment and agree in principle. But in the discussion, we wish to give readers a wider perspective regarding the plausible immune evasion mechanisms of HCMV, which may occur via interaction between ET-1 and dendritic cells since both the function and survival of dendritic cells have been suggested to be dependent on the endothelin axis (see REF 71). This should warrant further studies to look at this aspect.
Major issues
The higher MOI used here resulted in the infection of 100% of the cells. In vivo severe infections do not achieve 1% infection. Accordingly, there is a strong precedent in the mouse literature for conflicting effects in vivo & in vitro. Hence there is a need in an individual culture to distinguish between cells infected and cells affected by HCMV. At the least this should be discussed as a limitation.
A: We agree with the reviewer that a high MOI in vitro may not always reflect the situation in vivo. For this limitation and as suggested by the reviewer, we have added wording ‘in vivo’ to make this clearer in our revised ms (see line 380).
On this theme, it would also be helpful to see the immunohistochemistry at a higher magnification and with a clearer description of which cells were infected and affected. Were the infected cells altered morphologically? Eg: rounded up and non-functional by other assessments?
A: It is unclear to us if the reviewer here refers to infected and affected cells as the same cell, or if referring to cells affected by infected cells (i.e. bystander effects). We suspect the former. In vitro, HCMV usually causes enlargement of endothelial cells and formation of foci of infection before eventually leading to cell lysis at the later stages. The cytopathic effects of HCMV in fibroblast cells, MRC-5, on the other hand, tend to lead to lytic infection and cellular destruction. We have commented on this in the revised ms (lines 362-365). The immunofluorescence staining is used to corroborate our western blot’s results and may not be the optimal for assessing the morphological changes of the cells.
In the present study, we find evidence that HCMV IE expression affects ET-1 expression. It is possible that uninfected cells could be affected by paracrine effect of many different cytokines and growth factors produced or induced by HCMV infection in cells.
Higher magnified images are included in figure 4A-C in the revised ms.
B2M and 18S RNA seem to be used interchangeably to standardize the mRNA assays. Were levels of these 2 mRNAs directly related?
A: We found that both commonly used housekeeping genes B2M and 18S rRNA are minimally affected by HCMV infection. Thus, they can both be used for normalizing ET-1 expression. However, in the siRNA experiment, we found that 18S rRNA was more constant than B2M under the designated treatment. Hence, we therefore chose to use 18S rRNA for normalization of values in the siRNA experiment. We have added sentences ‘..where appropriate’ to make this clearer under the Materials and Methods section in the revised ms (see line 242).
The ELISAs were recorded as OD…. Can we be sure all measurements were made in the linear portion of the curve?
A: The ELISA test was performed and quantified using chemiluminescent reading and reported as relative light unit according to the vendor. A standard curve was used to make sure the quantification was within the linearity of the assay as instructed in the vendor’s protocol (R&D System) (line 348 and lines 248-253).
How was MOI determined? In what cell type?
A: We apologize for omitting this important information in our ms. The MOI was determined using a plaque assay in fibroblast cells (MRC-5). We have added this information in the revised ms (lines 168-172).
The statistical analysis is described simply as mean+SD. Is that based on replicate wells (how many) in a single experiment…or mean values derived from several experiments?
A: All experiments were performed in 2 or 3 experiments with 2-3 biological replicates. We have added the requested information in the Figure legends in the revised ms.
Reviewer 2
Comments and Suggestions for Authors
In the present paper, Koon-Chu Yaiw and Collegues, demonstrate that HCMV downregulates endotelin-1 (ET-1) expression and secretion, mainly via the action of its IE2-p86 104 protein, with a simultaneous accumulation of the ET-1 precursor protein preproET-1 in 105 infected cells. This phenomenon is jointly associated with an inhibition of ECE-1.
The work is indeed very interesting, since it suggests a possible mechanism of interference of the CMV with the endothelial system. However it needs minor improvements in the quality and description of some figures and in the text.
We thank the reviewer for this positive note on our study.
Revision Points:
1) Lane 131 when cells had reached a confluence of about 80-85% confluence in 12-well plates. Confluences is repeated, please change it.
A: We have corrected this typo error in the revised ms (see line 185).
2) Fig.2 (A and C) dpi value in A and C panel is missing. Why there’s no kinetics for this cell type? I think it’s not possible to compare this experiment to the previous without a similar kinetics. Maybe at least 2 points should be reported….
A: We apologize for omitting this information about dpi. We have added information about dpi in Figure 2 (A &C). Our focus in this study was HUVEC and hPASMC cells representing two cell types, i.e. endothelial cells and smooth muscle cells that are relevant cell types for the endothelin axis. We used another type of endothelial cells, hPAEC and of smooth muscle cells, hSMC, to corroborate our findings. Therefore, we limited the experiment to the indicated dpi, for confirmation of consistency of the viral effects on different cell types. Indeed, HCMV reduced ET-1 transcript levels in all examined cell types.
3) Fig3 you are describing a reduction of the release of ET-1. Thus to strengthen data obtained by real-time PCR I would suggest to show ET-1 modulation also by western blot analysis ( so the effect on the reduction of RNA corresponds to a reduction of the protein).
A: We concur with the reviewer that western blot analysis could have been good to include, but unfortunately, we have had difficulties to detect ET-1 peptide by western blot. The ET-1 peptide has a short half-life making western blot a daunting task, and the analysis may give unreliable results. However, we managed to detect the pre-proET-1 protein by western blot as shown in Figure 4D. Since most ET-1 peptide is secreted to supernatants, we found the chemiluminescent ELISA assay to be a more suitable test for the purpose.
4) Fig4 to better clarify the epitope recognized by specific ET-1 ab, it would be useful to add a schematic picture of the ET-1 aa sequence with the corresponding antigen binding site. ).
A: Unfortunately, this information is not provided by the manufacturer of the commercial antibody we used to detect the ET-1
5) Fig 4: why you analyzed IF with a different dpi respect to the real time PCR? Moreover, could you provide a better images or a higher magnification? ).
A: We agree this inconsistence is not optimal. We had intended to collect data of the same time course as in the qPCR. However, owing to unexpected events that occurred at the time when the experiments were performed, the sample collections were done a day later, i.e. 4dpi instead of 3dpi, and at 6dpi instead of 5dpi. Based on the increased ET-1 protein expression in all tested MOIs of virus infection coupled with the western blot findings, we are confident that experiment done at 3- and 5-dpi should produce similar findings at 4- and 6-dpi, respectively. The importance is to follow the infection over time, and we observe consistent results at early and late times after infection.
Higher magnified images are now included in figure 4 in revised ms.
6) Fig4. It’s not clear from the figure which kind of cells are infected. Please specify cell type on the figure. A: We apologies for the missing information, which we have now added to the legend of Fig 4.
7) Fig.4 D: it’s no clear why there are two lane for each sample. Please describe samples in the figure legend.
A: These double lanes are intended to demonstrate representative examples for 2 biological repeats, which demonstrate consistency of our findings. We have added this information in the Figure legend 4 in the revised ms.
8) Lane 233….”the ET-1 precursor protein levels of preproET-1 accumulated in endothelial cells infected at a high MOI (MOI 10).” The sentence is redundant, please change it.
A: We have amended this sentence (see lines 362-365) in the revised ms.
9) Line 234-235. The author state that ” We also noted that increase pre-proteinET-1 expression was associated with expression of the HCMV IE86 protein (Fig. 4D)”. According to my opinion, the amount of expressed HCMV IE86 protein seems to be similar to the expression of the two nearest lane (on the left), the only difference is the appearance of an upper band: is this an isoform of the same protein or a not specific product? Please clarify in the text or in the figure legend. Moreover, if you assert that there is and increasing expression of HCMV IE86 protein, you should quantify the band.
A: We agree with the reviewer that we have not illustrated this clearly in the Figure. We have now depicted both IE72 and IE86 in Fig. 4. In short, as pointed out by the reviewer, the expression of the lower band, i.e. IE72 appears to be similar, while the upper band IE86 is clearly increased with MOI=10 but hardly detectable in cells infected with an MOI of 0.1 and not detected in uninfected cells. The expression of IE86 seems to be dependent on the MOI; this is expected, and a quantification would be redundant (Figure 4D). However, we agree with the reviewer that quantification of both ECE-1 and IE86 should be quantified in Figure 4E, and we have added these data in the revised Figure 4E.
10) Fig.4 panel E: it is not reported the protein detected by wb in the middle panel, moreover the quality of typing is not clean (grey background).
A: We thank reviewer for bringing these points up. This was in part caused by the problem of copying the image to the document file for submission. We have further added the relevant information to the Figure and improved the image quality (see the revised Figure 4E).
11) Lane 240 author assert “We found that HCMV infection reduced ECE-1 expression levels in a dose-dependent manner (Fig. 4E)” Also in this case (see comment above) the reduced band should be quantified. In order to maintain the correct kinetic progression please horizontal flip the wb or the picture is misleading….
A: We have flipped the picture and quantified the expression of both IE86 and ECE-1 proteins (see revised Fig.4E).
12) Fig4. The figure legend should follow the title of the figure or it seems part of the text.
A: We have added the relevant information to all Figure legends as suggested by both reviewers (see the revised ms).
13) Lane 275, maybe in virus replicating cells is more appropriate sentence instead of cells replicating the virus.
A: We agree with the reviewer and have corrected this (see line 461).
14) Fig 7. As commented above, figure legend should follow the title of the figure or it seems part of the text
A: We thank the reviewer for bringing this up and have amended the text accordingly.
15) In all figures the p value is missing, please add it in the figure legends…..
A: We apologies for omitting this information and have added the p values and its definition in the revised ms (see also answer to reviewer 1).
16) line 332-333 “As ET-1 is a vasoconstrictive peptide, one would, under such scenario expect less vascular contraction and reduced blood pressure.” The comma is missed after “under such scenario”
A: We have added the comma as suggested (see line 580).
17) “Complicating the whole picture is the virus also affects the renin-angiotensin system that will also affect blood pressure levels”. This sentence is not grammatically correct I would suggest:” a further complication is also given by the fact that the virus also affect……”
A: Thank you for this suggestion. We have rephrased the se sentence in the revised ms (see lines 588-590).

Reviewer 2 Report
In the present paper, Koon-Chu Yaiw and Collegues, demonstrate that HCMV downregulates endotelin-1 (ET-1) expression and secretion, mainly via the action of its IE2-p86 104 protein, with a simultaneous accumulation of the ET-1 precursor protein preproET-1 in 105 infected cells. This phenomenon is jointly associated with an inhibition of ECE-1.
The work is indeed very interesting, since it suggests a possible mechanism of interference of the CMV with the endothelial system. However it needs minor improvements in the quality and description of some figures and in the text.
Revision Points:
1) Lane 131 when cells had reached a confluence of about 80-85% confluence in 12-well plates. Confluences is repeated, please change it.
2) Fig.2 (A and C) dpi value in A and C panel is missing. Why there’s no kinetics for this cell type? I think it’s not possible to compare this experiment to the previous without a similar kinetics. Maybe at least 2 points should be reported….
3) Fig3 you are describing a reduction of the release of ET-1. Thus to strengthen data obtained by real-time PCR I would suggest to show ET-1 modulation also by western blot analysis ( so the effect on the reduction of RNA corresponds to a reduction of the protein)
4) Fig4 to better clarify the epitope recognized by specific ET-1 ab, it would be useful to add a schematic picture of the ET-1 aa sequence with the corresponding antigen binding site.
5) Fig 4: why you analyzed IF with a different dpi respect to the real time PCR? Moreover, could you provide a better images or a higher magnification?
6) Fig4. It’s not clear from the figure which kind of cells are infected. Please specify cell type on the figure.
7) Fig.4 D: it’s no clear why there are two lane for each sample. Please describe samples in the figure legend.
8) Lane 233….”the ET-1 precursor protein levels of preproET-1 accumulated in endothelial cells infected at a high MOI (MOI 10).” The sentence is redundant, please change it.
9) Line 234-235. The author state that ” We also noted that increase pre-proteinET-1 expression was associated with expression of the HCMV IE86 protein (Fig. 4D)”. According to my opinion, the amount of expressed HCMV IE86 protein seems to be similar to the expression of the two nearest lane (on the left), the only difference is the appearance of an upper band: is this an isoform of the same protein or a not specific product? Please clarify in the text or in the figure legend. Moreover, if you assert that there is and increasing expression of HCMV IE86 protein, you should quantify the band.
10) Fig.4 panel E: it is not reported the protein detected by wb in the middle panel, moreover the quality of typing is not clean (grey background).
11) Lane 240 author assert “We found that HCMV infection reduced ECE-1 expression levels in a dose-dependent manner (Fig. 4E)” Also in this case (see comment above) the reduced band should be quantified. In order to maintain the correct kinetic progression please horizontal flip the wb or the picture is misleading….
12) Fig4. The figure legend should follow the title of the figure or it seems part of the text.
13) Lane 275, maybe in virus replicating cells is more appropriate sentence instead of cells replicating the virus.
14) Fig 7. As commented above, figure legend should follow the title of the figure or it seems part of the text.
15) In all figures the p value is missing, please add it in the figure legends…..
16) line 332-333 “As ET-1 is a vasoconstrictive peptide, one would, under such scenario expect less vascular contraction and reduced blood pressure.” The comma is missed after “under such scenario”
17) “Complicating the whole picture is the virus also affects the renin-angiotensin system that will also affect blood pressure levels”. This sentence is not grammatically correct I would suggest:” a further complication is also given by the fact that the virus also affect……”
Author Response

(The authors gave the same response as above.)

Round 2
Reviewer 1 Report
I used the term "affected by CMV" to describe cells that were not productively infected. You have strictly failed to address this point. Nonetheless the effects described were clearest at a MOI that ensured that most cells were infected. A potential mechanism for the immunomodulatory effects of CMV is therefore revealed.